# Single-Cell RNA Sequencing Reveals Differences in Chromatin Remodeling and Energy Metabolism among In Vivo-Developed, In Vitro-Fertilized, and Parthenogenetically Activated Embryos from the Oocyte to 8-Cell Stages in Pigs

**DOI:** 10.3390/ani14030465

**Published:** 2024-01-31

**Authors:** Jianlin Fan, Chang Liu, Yunjing Zhao, Qianqian Xu, Zhi Yin, Zhonghua Liu, Yanshuang Mu

**Affiliations:** 1Key Laboratory of Animal Cellular and Genetic Engineering of Heilongjiang Province, Northeast Agricultural University, Harbin 150030, China; vanqilei@163.com (J.F.); s210901071@neau.edu.cn (C.L.); j18845832622@163.com (Y.Z.); xuqianqian@neau.edu.cn (Q.X.); zyin@neau.edu.cn (Z.Y.); 2College of Life Science, Northeast Agricultural University, Harbin 150030, China

**Keywords:** pig, in vivo embryo, IVF embryo, parthenogenetic activation embryo, single cell sequence

## Abstract

**Simple Summary:**

This study investigated the developmental differences between in vivo-developed (IVV) and in vitro embryos (in vitro fertilization (IVF) and parthenogenetic activation (PA)) in pigs using single-cell RNA sequencing data from the oocyte to the 8-cell stages. Findings show that in vitro embryos have more similar developmental trajectories and lower gene diversity, especially PA, compared to IVV. Significant variations were observed in maternal mRNA, influencing mRNA splicing, energy metabolism, and chromatin remodeling, where key genes like *SMARCB1* and *HDAC1* (in vivo) and *SIRT1* and *EZH2* (in vitro) might be playing crucial roles. Despite similar ZGA timings across embryo types, IVV embryos exhibited more upregulated genes, particularly during major ZGA, and global epigenetic modification differences were evident from the oocyte stage and continued to expand during minor ZGA. They also uniquely upregulated genes linked to mitochondrial function, ATP synthesis, and oxidative phosphorylation. A notable difference in mRNA degradation between in vivo and in vitro embryos was observed during major ZGA, and the timely decay of maternal mRna in IVV except in in vitro embryos was related to phosphatase activity and cell junctions. The study also highlighted the higher expression of mitochondrially encoded genes in in vitro embryos, though nucleosome occupancy and *ATP8* expression were higher in IVV.

**Abstract:**

In vitro-fertilized (IVF) and parthenogenetically activated (PA) embryos, key to genetic engineering, face more developmental challenges than in vivo-developed embryos (IVV). We analyzed single-cell RNA-seq data from the oocyte to eight-cell stages in IVV, IVF, and PA porcine embryos, focusing on developmental differences during early zygotic genome activation (ZGA), a vital stage for embryonic development. (1) Our findings reveal that in vitro embryos (IVF and PA) exhibit more similar developmental trajectories compared to IVV embryos, with PA embryos showing the least gene diversity at each stage. (2) Significant differences in maternal mRNA, particularly affecting mRNA splicing, energy metabolism, and chromatin remodeling, were observed. Key genes like *SMARCB1* (in vivo) and *SIRT1* (in vitro) played major roles, with *HDAC1* (in vivo) and *EZH2* (in vitro) likely central in their complexes. (3) Across different types of embryos, there was minimal overlap in gene upregulation during ZGA, with IVV embryos demonstrating more pronounced upregulation. During minor ZGA, global epigenetic modification patterns diverged and expanded further. Specifically, in IVV, genes, especially those linked to H4 acetylation and H2 ubiquitination, were more actively regulated compared to PA embryos, which showed an increase in H3 methylation. Additionally, both types displayed a distinction in DNA methylation. During major ZGA, IVV distinctively upregulated genes related to mitochondrial regulation, ATP synthesis, and oxidative phosphorylation. (4) Furthermore, disparities in mRNA degradation-related genes between in vivo and in vitro embryos were more pronounced during major ZGA. In IVV, there was significant maternal mRNA degradation. Maternal genes regulating phosphatase activity and cell junctions, highly expressed in both in vivo and in vitro embryos, were degraded in IVV in a timely manner but not in in vitro embryos. (5) Our analysis also highlighted a higher expression of many mitochondrially encoded genes in in vitro embryos, yet their nucleosome occupancy and the *ATP8* expression were notably higher in IVV.

## 1. Introduction

Pigs, as mammals, play a vital role in agriculture and possess numerous attributes that make them superior models for translational and clinical research; their anatomical size and structure, immunology, genome, and physiology closely resemble those of humans [1]. The study of early embryo development in pigs is of immense significance, drawing keen interest from scientists in genetics, clinical research, developmental biology, and reproductive biology [2,3,4].

Mammalian embryo development commences with the fertilization of an oocyte by a sperm; however, the natural processes leading to mammalian embryo development are diverse and varied. In the context of pig reproduction, advanced reproductive technologies such as in vitro fertilization (IVF), parthenogenetic activation (PA), oocyte in vitro maturation (IVM), and in vitro embryo culture have been devised and applied [5]. These methods lead to the development of embryos that undergo multiple rounds of cleavage within a controlled laboratory environment, eventually reaching the blastocyst stage. While genome editing and somatic cell nuclear transfer techniques employed for genetic engineering purposes necessitate the use of these in vitro embryos, the PA embryo holds the advantage of stably preserving the dominant genotype [6,7]. However, when compared to embryos resulting from natural in vivo development (IVV), IVF and PA embryos exhibit noticeable phenotypic defects during embryonic development, including irregular cell morphology, slower developmental cleavage rates, lower rates of embryonic formation, lower pregnancy rate after transfer, and small litter size [8,9,10]. PA embryos that lack paternal genomes even face embryonic mortality [6].

Following fertilization in mammals, a critical process known as the maternal to zygotic transition (MZT) takes place. During the MZT, there are two major molecular events—maternal clearance and zygotic genome activation (ZGA)—which are highly correlated and transfer the developmental control from maternal mRNA to the zygotic genome together [11,12]. A timely mRNA clearance is essential for the proper initiation of ZGA, and the proper initiation of ZGA is also essential for the timely degradation of mRNA [13,14]. During these events, embryos undergo significant epigenetic reprogramming, including DNA and histone methylation, histone acetylation, and remodeling of chromatin 3D structure regulated by large chromatin remodeling complexes, contributing significantly to dynamic gene expression while transforming oocytes and sperms into totipotent embryos and ensuring subsequent development [15,16,17].

ZGA in mammals follows two distinct waves known as minor and major ZGA, with minor ZGA essential for subsequent gene activation [18], but the timing and nature of these events differ among species. In mice, minor ZGA initiates at the one-cell stage, followed by major ZGA at the two-cell stage [12]. In contrast, pigs and humans share a similar timing for minor ZGA, while their major ZGA events more closely resemble each other, occurring around the four-cell to eight-cell stages [19,20,21].

The transcription initiation of ZGA in early embryonic development depends on the presence and levels of key transcriptional regulators. Once these factors reach a certain threshold level, transcriptional activation occurs [12], and the absence of transcription factors specific to certain genes results in transcriptional silencing [22]. *DUX* was the initial transcription factor responsible for initiating transcription both in early mouse and human embryos [23]. *POU5F1*, *SOX2*, *c-MYC*, and *KLF4* successfully reprogrammed mature cells into pluripotent stem cells [24]. These factors, along with *NANOG*, play a significant role in zygotic genome activation [25,26,27]; among them, the relationship between *SOX2* and *POU5F1* is particularly close and interdependent [28,29].

The various cellular processes in ZGA, such as the chromatin reprogramming mentioned above and biological synthesis, characterize the initial stages of preimplantation development in mammals as a highly demanding process for energy, necessitating substantial quantities of ATP [30,31], which is produced exclusively by the approximately 100,000 inner mitochondria [32] that are often referred to as the ‘powerhouses of the cell’. Their major role in supplying the necessary energy is driven by both mitochondrial and nuclear DNA; therefore, it is evident that mitochondrial genes must play a crucial role in orchestrating and supporting the intricacies of embryogenesis [33]. While the types of metabolic substrates used by mammals during early development are generally similar across different species, their reliance on these substrates can vary. Pyruvate stands out as a crucial metabolic substrate essential for ATP production in humans and mice during early development, while lipids, especially in porcine and bovine embryos, occupy the role that pyruvate plays in other species as a major energy source in this critical process [32].

In this study, to investigate and compare the differences in MTZ from porcine IVV embryos and those generated in vitro using assisted reproductive techniques, we integrated and analyzed single-cell transcriptomic data for IVV, IVF, and PA embryos at the oocyte to eight-cell stages. In our analysis, we aimed to provide insights regarding ZGA functional differences and the molecular mechanisms governing these differences, shedding light on the impact of assisted reproductive techniques on in vitro early embryo development.

## 2. Materials and Methods

### 2.1. Ethics Statement

All animal-related procedures followed the guidelines set forth by the Institutional Animal Care and Use Committees (IACUCs) of Northeast Agriculture University (NO. NEAUEC20190118).

### 2.2. In Vivo Embryo Collection and Single Cell Separation

Embryos in our study were collected from the same batch of Bama sows of similar and appropriate mating ages (Appendix A). Following two rounds of mating (12 h interval), uteri were obtained, and embryos were flushed out at specific developmental time points. Oocytes were collected 24 h after estrus; 1-cell-stage embryos were collected 19 h after mating, with visible sperm on the zona pellucida and second polar body within the perivitelline space; 2-cell-stage embryos were collected 40 h after mating; 4-cell-stage embryos were collected 60–70 h after mating; 8-cell-stage embryos were collected 3.5–4 days after mating.

The flushed embryos were observed under a microscope, and blastomeres of a uniform size and corresponding developmental stage were selected. Embryos with sperm on the zona pellucida were placed in a carbon dioxide incubator for a 2 h recovery period. After this, single blastomere isolation was performed in two steps.

First, ICM and TE were separated. Blastocysts with visible blastocyst cavities at 5.5 days were placed into an operating solution, with the ICM-like side of the embryo aspirated using a fixed needle. An Eppendorf FemtoJet injection system was used to inject 0.25% trypsin along the 3 o’clock direction until a mass of cells was slowly detached from the blastocyst and the zona pellucida was eventually expelled. The zona pellucida was then removed using an acidic operating solution, and the cells were gently blown using a thin glass needle (mouth pipette), yielding two cell clusters—larger cells were TE, and smaller cells were ICM. After washing the clusters three times with DPBS, they were placed in a microcentrifuge tube containing lysis buffer and stored at −80 °C for later use.

Second, single-blastomere isolation was performed. Embryos at each stage were treated with an acidic operating solution to remove the zona pellucida. They were then washed twice with calcium- and magnesium-free DPBS before being placed in accutase solution for approximately 30 min. Using a mouth pipette with a diameter appropriate for the size of a single blastomere, the embryos were gently blown and aspirated to separate individual blastomeres. These single blastomeres were then washed three times with DPBS and placed in a microcentrifuge tube containing lysis buffer before being stored at −80 °C for later use.

### 2.3. Single-Cell Transcriptome Amplification and RNA Sequencing 

Smart-Seq2 method was used to amplify the single-cell sample. The reaction system for the first cDNA strand synthesis comprised the reverse transcription enzyme, buffer, template-switching oligo (TSO) primers, and oligo-dT primers with common sequence. For second-strand synthesis, the PCR amplification reagent and ISPCR primer with common sequence were added to the first-strand synthesis products. The sample cDNA was fragmented into 300 bp using a Bioruptor^®^Sonication System (Diagenode Inc., Denville, NJ, USA), and then terminal repair, addition of A, ligation of sequencing adapters, and amplification were performed. The PCR amplification product, which was of 350–450 bp, was extracted. Then, the concentration was measured using a Qubit 2.0 Flurometer (Life Technologies, Carlsbad, CA, USA). The integrity of the amplified products was detected using an Agilent 2100 High-Sensitivity DNA Assay Kit (Agilent Technologies, Santa Clara, CA, USA). The libraries were sequenced on an Illumina HiSeq X Ten platform, employing 150 bp paired-end sequencing.

### 2.4. Data Acquisition and Processing

Raw read data for IVF (in vitro fertilization) and PA (parthenogenetic activation) from 1–8-cell stages were obtained from the NCBI GEO repository (http://www.ncbi.nlm.nih.gov/geo/ (accessed on 19 July 2023)) under accession number GSE164812. Data for in vitro matured oocytes were acquired from NCBI GEO under accession number GSE160334. The raw reads were initially processed using fastp (version 0.23.2) [34] to trim adapters and filter out reads with low quality. The resulting high-quality reads were then aligned to the pig reference genome (Sscrofa11.1) using the STAR aligner (version 2.7.10a) [35]. Gene expression levels (count, TPM (Transcripts Per Million), FPKM (Fragments Per Kilobase of transcript per Million mapped reads)) were quantified using Rsem (version 1.3.3) [36] with default settings. 

### 2.5. Preliminary and Differential Analyses

TPM values from three different subsets (IVV, IVF, PA) were analyzed using the Seurat package (version 4.3.0) [37] for preliminary analysis. Pseudotime progression was inferred using Monocle3 (version 1.3.1) [38]. Differentially expressed genes between any two groups were identified using DESeq2 (version 1.38.3) [39] based on counts. However, for groups with fewer than three samples, EdgeR (version 3.40.2) [40] was employed as an alternative. A gene was considered significant if the Benjamini and Hochberg-adjusted *p*-value (Padj.) was <0.05 and the |log2(Fold change)| ≥ 1.

### 2.6. Enrichment Analysis

Gene enrichment analysis, including Gene Ontology (GO) and KEGG pathways, was conducted after converting pig genes to their human orthologs using the ClusterProfiler package (version 4.6.2) [41] under org.Hs.eg.db package (version 3.16.0) for better annotation data, with a significance threshold of FDR < 0.05. Enrichment results were further clustered and simplified using the simplifyEnrichment package (version 1.8.0) [42]. Gene set enrichment scores were obtained using the GSVA package (version 1.46.0) [43] and analyzed with ClusterProfiler. For motif enrichment and prediction of upstream transcription factors, the RcisTarget package (version 1.18.2) [44] was utilized.

### 2.7. Protein–Protein Interaction Analysis

Protein–protein interactions (PPI) were investigated using the STRING database (https://string-db.org/ (accessed on 10 September 2023)) and visualized in Cytoscape (version 3.10.1) [45].

## 3. Results

### 3.1. Sample Selection and Comprehensive Characteristic Analysis of RNA Sequencing in IVV, IVF, and PA Embryos

To investigate the molecular mechanisms underlying the differences in early embryonic development among IVV, IVF, and PA embryos, we integrated and analyzed single-cell transcriptomic data from the oocytes to the eight-cell stage for IVV, IVF, and PA embryos (IVV starts from in vivo matured oocytes, while PA and IVF start from in vitro matured oocytes). Pseudotemporal and correlation analyses were used to exclude samples falling into adjacent developmental stages, aiming to improve the stage specificity, such as removing three IVF samples from the four-cell stage that were located in the eight-cell stage in subsequent analyses (Figure 1B and Appendix A).

The characterization differentiation of the overall transcriptome data for these three types of embryos was performed through hierarchical clustering and genome-wide TPM-based 3D UMAP with pseudotemporal trajectories. The figure reveals two distinct developmental trajectories, separating IVV from in vitro embryos that share the same trajectory (Figure 1B). Within each cluster and trajectory, oocytes and 1–2-cell stages were consistently closer, while the 4- and 8-cell stages were clearly distinct from other stages, consistent with the major ZGA (Figure 1A,B). The divergence between IVV and in vitro embryos starts at the very beginning, at the oocyte stage, and becomes significantly larger at the four-cell stage, suggesting abnormalities in major ZGA for in vitro embryos (Figure 1B).

Heatmaps from hierarchical clustering vividly displayed the presence of stage-specific genes, with a significant number of 4- and 8-cell stage-specific genes not being transcribed in in vitro embryos compared to in vivo embryos (Figure 1A). Moreover, stage-specific genes of in vivo and in vitro matured oocytes (i.e., maternal mRNAs) not only showed clear differences but also exhibited varied patterns of degradation and stability in the 4–8-cell stages between in vivo and in vitro embryos (Figure 1A).

Further exploring the expression characteristics, we created box plots of the total gene expression, the number of different genes, and the percentage of mitochondrial (MT) gene expression for each sample (Figure 1C). The results showed an increase in mitochondrial gene expression and a decrease in gene variety across the 1–8-cell stages (Figure 1C). Compared to the sudden increase and heterogeneity of in vitro embryos at the four-cell stage, the MT percentage in in vivo embryos was more stable (Figure 1C). In vitro embryos generally exhibited a higher total gene expression across stages, while the number of genes during the 1–8-cell stages (potentially reflecting developmental status and potential) was generally lower, with PA embryos showing the least gene diversity (Figure 1C).

Finally, we explored the comparative expression trajectories of five common pluripotency factors that are important in early embryo development—*POU5F1*, *SOX2*, *KLF4*, *NANOG*, and *MYC*—during the 1–8-cell stages in these three types of embryos (Figure 1D). The latter three factors began to be notably upregulated from the 2–4-cell stage across all embryos, while *POU5F1* and *SOX2* showed significant differences between in vivo and in vitro conditions (Figure 1D). *POU5F1* maintained a stable high expression throughout the 1–8-cell stages in vivo, whereas it was significantly upregulated only at the 2–4-cell stages in vitro and did not persist into the subsequent stages; in vitro embryos exhibited deficiencies in this aspect (Figure 1D). Additionally, *SOX2*, with a tight association with *POU5F1*, exhibited continuous upregulation from the 2–4 cell stage in vivo, while it remained largely unexpressed in vitro (Figure 1D).

### 3.2. Differential Expression of Maternal mRNA in In Vivo and In Vitro Matured Oocytes

From our initial results, we can infer that at the embryonic oocyte–8-cell stages’ developmental starting point, significant differential mRNAs are present between oocytes matured in vivo and in vitro, upon which early embryo ZGA depends. This disparity may directly influence the adjacent one- and two-cell stages (Figure 1A,B). Consequently, we conducted a differential analysis of the oocytes matured in vivo over in vitro, confirming significant upregulated genes with a threshold: FDR < 0.05, log2FC > 1, and classified genes with FPKM > 10 as respective maternal mRNAs based on empirical partitioning [20] (Figure 2A). The heatmap of maternal mRNA, segmented based on the significance of gene expression differences and whether genes are shared or unique, shows that the majority of abundant maternal mRNAs are common to both types (Figure 2C). However, marked variations were predominantly found in the significantly expressed genes (Figure 2C). Notably, even though not significant, there were still 785 unique genes in the maternal mRNA pool of in vivo oocytes and 301 unique genes in in vitro oocytes (Figure 2A,C).

Compared to in vitro oocytes, in vivo matured oocytes demonstrated a higher number of significantly upregulated genes with substantial fold changes (Figure 2B). Remarkably, in vitro matured oocytes exhibited an extraordinarily high-fold change (log2FC approximately −20) in 35 genes, primarily associated with various stresses, metabolic processes, and signaling pathways (Figure 2B). In contrast, in vivo matured oocytes exhibited equal levels of fold change (log2FC approximately 20) in only six genes (Figure 2B). However, it is important to note that the GO/KEGG enrichment analyses and protein–protein interaction (PPI) studies did not yield significant associations for these 35 genes with other maternal genes or among themselves, suggesting that their erratic expression patterns influenced by environmental factors may be random aberrations.

Following zygote formation, oocytes experience substantial chromatin remodeling, an event that is crucial for shaping subsequent embryonic expression. Therefore, we initially explored some of the common chromatin remodeling-related genes in the two types of oocytes, discovering significant differences between them. Notably, *HDAC1* and *SMARCB1* were significantly upregulated in in vivo oocytes, although *SMARCA5*, *SMARCC1*, and *HDAC2* were significantly expressed in the opposite oocytes (Figure 2D). Additionally, we checked the expression levels of the five pluripotency factors we mentioned before that are actively involved in the reprogramming process of early embryos, among which only *POU5F1* had effective expression (as a maternal mRNA) and was validated to be significantly upregulated in in vivo oocytes (Figure 1D and Figure 2D). GO enrichment analysis indicated that *POU5F1* is involved in the maintenance of early embryonic stemness, cell division, and potential RNA-mediated gene silencing (Figure 2E).

To explore global functional differences, we conducted GO enrichment analyses followed by similarity clustering for the maternal mRNA sets of both oocytes (due to limited pig data, we converted the data to human orthologous genes for all enrichment analyses). Subsequently, we selected terms in which significantly expressed maternal genes in vivo/in vitro were involved for further analysis. These GO terms were found to be involved in various biological processes, including the cell cycle, cellular localization, cell stemness, gene expression regulation, and signaling pathways (Figure 3A and Appendix A). Notably, among the clusters of GO terms, category C in vivo and category B in vitro encompassed a variety of chromatin morphology-related modifications and mRNA processing, which are related to the regulation of gene expression (Figure 3A and Appendix A). These categories ranked first in both the number of terms and participating genes among all categories, indicating their significant contribution to the observed differences (Figure 3A and Appendix A). We found that the types of histone modifications that the differential genes were involved in between the two categories were largely the same, and the in vivo matured oocytes, compared to the in vitro, showed a predominance of H3-K14 and H2A acetylation, as well as H2B ubiquitination. Similarly, many differentially expressed genes were also distributed in the direction of histone modification regulation and DNA methylation. It was observed that in vivo matured oocytes had a higher abundance of histone modification regulation compared to in vitro matured oocytes, especially on the methylation of H3K4, which is generally related to gene activation, while the latter demonstrated a stronger upregulation capability in DNA methylation-dependent heterochromatin formation (Appendix A). Although no enrichment results were obtained for epigenetic regulation, a further gene set enrichment analysis (GSEA) confirmed significant energy-related differences in the KEGG pathways with a threshold of NES > 1 and FDR < 0.25, particularly in lipid metabolism (Figure 3D). It is noteworthy that lipid metabolism, amino acid metabolism, and glycolysis involve the production of acetyl-CoA, which is the direct material for epigenetic acetylation regulation. Additionally, *SMARCB1* in the large chromatin remodeling complex also requires ATP for its operation.

To identify possible hub genes, we created ‘remod’ gene sets by extracting genes associated with chromatin remodeling from GO terms in Cluster C for in vivo and Cluster B for in vitro. The genes were then screened for possibly significant involvement in these GO terms, characterized by their frequencies of participation in the related GO terms that exceeded a predefined threshold (Figure 3C). In vivo, the genes included *SMARCB1*, *HCFC1*, *IWS1*, *NFYC*, *EHMT2*, *MACROH2A1*, and *SNW1*; similarly, in vitro, the genes included *SIRT1*, *KDM1A*, *SMARCA5*, *RIF1*, *ZMPSTE24*, and *SETDB1*, all of which exhibited substantial differences in fold change and expression levels (Figure 3E and Appendix A). It is noteworthy that *SMARCB1* in the in vivo group and *SIRT1* in the in vitro group ranked first. *SMARCB1*, also known as *SNF5*, is a critical component of the SWI/SNF chromatin remodeling complex, and SIRT1 is a histone deacetylase that plays a central role in the control of histone acetylation levels, particularly H3K9 deacetylation.

Given that genes involved in chromatin remodeling typically function within larger complexes, we conducted a protein–protein interaction (PPI) analysis on the ‘remod’ gene sets to identify key players with the highest interaction degrees, highlighting their pivotal roles in these complexes (see Figure 3F and Appendix A). In vivo, the top five genes include *HDAC1* (with an involvement frequency of 3), *KAT2A* (frequency 13), *SETD1A* (frequency 8), *DNMT3A* (frequency 5), and *MLLT3* (frequency 4). In vitro, the top genes comprised *EZH2* (frequency 8), *KDM1A* (frequency 17), *SIRT1* (frequency 23), *HDAC2* (frequency 3), and *SUZ12* (frequency 4). Notably, *HDAC1*, ranking highest in the in vivo set, is primarily involved in histone deacetylation, whereas *EZH2*, ranking highest in the in vitro set, plays a crucial role in histone methylation, particularly in the trimethylation of H3K27.

Next, we validated the dynamic differences in the gene expression trajectories of these two ‘remod’ gene sets across subsequent developmental stages among embryos (Appendix A). We observed that although some of the significantly expressed genes from in vivo/in vitro oocytes experienced partial reversion in embryos produced from the other type of oocytes during minor ZGA; overall, significant differences were maintained (Appendix A). Furthermore, the expression levels of these genes exhibited a consistent decline from the oocyte to the eight-cell stage, suggesting their predominant roles in the early stages post zygote formation (Appendix A).

Finally, following the analysis of the ‘remod’ gene sets and their roles in chromatin remodeling, it is also important to consider mRNA splicing, which is correlated with gene expression. In vivo genes predominantly encompass components essential for basic splicing mechanisms and cellular functionality, such as *SNRPE*, *SNRPF*, and *U2AF1* (Appendix A). In contrast, in vitro genes, including specific splicing factors such as *SRSF1* and *SRSF3*, reflect a complex regulatory environment, indicating adaptation to specific in vitro conditions and developmental stages (Appendix A).

### 3.3. Functions of Differential ZGA Gene across Oocyte to Eight-Cell Stages among Embryos

In our analysis of mRNA upregulation dynamics from oocyte to eight-cell stages, we found in all three embryos that significantly upregulated genes (threshold: FDR < 0.05, log2FC > 1) were limited during the minor ZGA period (oocyte–1-cell and 1–2-cell stages), with fewer changes observed in the 1–2 cell stage compared to the oocyte–1-cell stage (Figure 4A). However, a notable increase in upregulated genes occurred during the 4–8-cell stage, marking the major ZGA period. These results indicate a consistent timing of both minor and major ZGA across all embryos, though based only on gene quantity (Figure 4A). As expected, IVV exhibited more upregulated genes at each stage than in vitro embryos, while PA showed a higher count than IVF in earlier stages (oocyte–4-cells), which was then exceeded in the 4–8-cell stage (Figure 4A). Remarkably, Venn diagrams demonstrated minimal overlap between gene sets of adjacent stages among embryos, and in conjunction with former results, differential chromatin remodeling genes within the maternal mRNA environment could be responsible for the observed patterns in minor ZGA (Figure 4B). Thus, using the GTRD database, we extracted genes significantly bound at their TSS regions by in vivo activation-related groups (*SMARCB1*, *HCFC1*, *KAT2A*, *SETD1A*) and in vitro silencing-associated groups (*EZH2*, *SUZ12*, *SETD1B*, *SIRT1*), which showed strong associations within each group, to confirm regulatory connections between core genes as previously noted and minor ZGA genes. These sets of genes correlated well with the set of nearly 900 genes independently upregulated during the first minor ZGA in IVV embryos (Appendix A).

In the premajor ZGA phase, maternal mRNA predominantly controls gene regulation, augmented by chromatin remodeling and the activation of both general and sequence-specific transcription factors, setting the stage for major ZGA. To discern functional differences in minor ZGA in collaboration with maternal mRNa that lead to minimal overlap in major ZGA gene sets, we conducted a GO enrichment analysis, which integrated maternal mRNA environments with genes significantly upregulated during minor ZGA as an enrichment environment for each embryo type—IVV, IVF, and PA. Despite further extraction by distinct minor ZGA gene sets for each type, the analysis revealed a consistent directionality in GO terms across all embryos, with significant enrichment particularly involved in areas of mRNA processing and chromatin remodeling, both critical for gene expression regulation (Figure 4C and Appendix A). This pattern indicates not only similar rough developmental trajectories but also broader underlying differences during minor ZGA, with IVV embryos exhibiting a markedly more diverse enrichment profile.

Further exploration in the comparative analysis of histone modifications and DNA methylation related to gene regulation among the three embryo types indicated further divergence in gene regulation during minor ZGA. Notably, compared to maternal mRNA, IVV exhibited a greater number and diversity of genes involved in histone modification patterns in H2A ubiquitination and H4 acetylation, outperforming both IVF and PA. However, PA showed better results than IVV and IVF in the methylation of H3-K9, H3-K4, and H3-K27 and in the acetylation of H3-K14 (Appendix A). Although these patterns had previously shown in vivo advantages over in vitro advantages during the oocyte stage, further confirmation found that the intersecting parts were still minimal (Appendix A). PA showed more genes regulating DNA methylation than IVF. These findings may partly explain the higher number of upregulated genes in PA than IVF during 2–4 major ZGA (Appendix A).

A comparative KEGG enrichment analysis during minor ZGA revealed IVV’s upregulation of basal transcription factors and genes associated with energy metabolism, implying an early reliance on lipids as an energy source in pig embryos and deficiencies in PA and IVF embryos in this regard (Figure 4D). It is worth noting that lipid metabolism is closely related to acetylation, with acetyl-CoA serving as an essential intermediate. The AMPK signaling pathway, which is essential for regulating cellular energy balance, displayed varied expression patterns (Figure 4E). In IVV embryos, upregulated AMPK complex subunits *PRKAG1*, *PRKAG2*, and *PRKAA1* showed a continual increase, whereas in vitro embryos, especially PA, demonstrated minimal expression of *PRKAG1* and *PRKAG2*, pointing to potential energy-sensing and -regulation issues (Figure 4E).

To further confirm global differences in chromatin remodeling during minor ZGA, we conducted a gene set variation analysis (GSVA) among the three embryo types from the oocyte to the two-cell stage (Figure 4F and Appendix A). The results indicated that during minor ZGA, DNA methylation-driven heterochromatization processes in in vitro embryos were more pronounced than those in in vivo embryos (Figure 4F). This difference appears to be directly inherited from the maternal mRNA of oocyte stages because the degree of DNA methylation involvement from starting at the oocyte to the end of the two-cell stage was similar among all three embryos (Figure 4F). In contrast, in vivo embryos exhibited stronger histone acetylation, especially histone H3 acetylation, during the same stages, while in vitro embryos demonstrated higher activity in deacetylation processes (Appendix A).

To expand our investigation on the contribution of sequence-specific transcription factors to major ZGA, significantly upregulated genes of the 2–4 cell stage were subjected to a motif-enrichment analysis. The top-scoring motifs *ELK1*, *ELK3*, *ELK4*, and *GABPA* were selected for a comparative analysis of their expression patterns in the three types of embryos (Appendix A). We found no comprehensive differences in the expression patterns of these transcription factors during minor ZGA among the three embryos (Appendix A).

After the two-cell stage, major ZGA became the focus in the MZT, accompanied by significant changes in the transcriptome. To delve into the functional alterations and differences among the three embryo types during this phase, we conducted enrichment analyses on the independent subsets of genes that were significantly upregulated during the major ZGA in each embryo type (Figure 5A–E and Appendix A). Specifically, during the 2–4-cell stage of major ZGA, IVV exhibited a rich and diverse profile in its independently enriched GO and KEGG terms, with a significant focus on various energy metabolism processes (Figure 5A,D). These processes include the assembly of mitochondrial electron transport chain complexes, regulation of mitochondrial genes, ATP synthesis, and several histone acetylation processes related to chromatin remodeling, potentially setting the stage for the next phase of major ZGA (Figure 5C). For IVF, the focus was more on sensing external stimuli, substance transport, and cell-cycle regulation, while PA showed a greater emphasis on epigenetic modifications (Appendix A).

At the 4–8-cell stage of major ZGA, in vitro embryos showed almost no significant enrichment, whereas IVV displayed a richer array of terms, particularly in molecule degradation, metabolism, and energy production (Figure 5B). KEGG analysis indicated a more pronounced utilization of lipid pathways, such as the PPAR pathway and fatty acid metabolism, in IVV embryos (Figure 5E). Notably, some genes maintained upregulation across both major ZGA stages (Figure 1A). By clustering, we identified this set of genes related to mitochondrial functions and ATP synthesis (Appendix A). This highlights the growing disparity in energy metabolism regulation between in vivo and in vitro embryos during the 4–8-cell stage of major ZGA, emphasizing the importance of energy metabolism (Appendix A).

### 3.4. Different Maternal mRNA Decays of Adjacent Stages from the Oocyte to the 8-Cell Stage

Another critical process during the MTZ is the decay of maternal mRNA, which is intimately related to the regulation of early embryonic gene expression. Our analysis of significantly downregulated mRNAs at various stages in IVV, IVF, and PA embryos revealed that mRNA degradation begins from the zygote stage in all embryos, with a notable increase at the 4-cell stage and peaking at the 8-cell stage in in vivo embryos. This peak is significantly higher than any stage in the in vitro embryos, where fewer genes are degraded at each stage.

Next, we compared the common set of maternal mRNA with nonsignificant differences between in vivo and in vitro embryos. We observed that maternal mRNA degradation in in vitro embryos is largely incomplete, potentially due to differences in genes associated with maternal mRNA degradation (Figure 6A). A heatmap of maternal mRNA degradation-related genes revealed distinct categorizations into two groups at the minor and major ZGA stages, showing marked differences between in vivo and in vitro embryos at every stage (Figure 6B).

During the minor ZGA stage, some genes in IVV embryos were significantly higher than those in in vitro embryos, mainly involving mRNA binding. This difference became even more pronounced during the major ZGA stage. The extreme incompleteness of ZGA in in vitro embryos resulted in the complete overshadowing of their related gene expression levels by IVV (Figure 6B,C).

From minor to major ZGA, genes that specifically bind to mRNA and recruit mRNA degradation complexes transit from *CPEB1*, *KHSRP*, *ZFP36L1*, *ZFP36L2*, and *AGO2* to *YTHDF2*, *ZFP36*, and others involved in mRNA degradation, processing a shift from the death complex formed by *TENT4A*, *TENT4B*, and *PAN3* to *CNOT1/2/3/4/6/8/9/10*, *DCP1A/DCP1B/DCP2*, *EXOSC3/4/5/9/10*, and *XRN1*, while *CNOT7* is highlighted as a core interaction protein for maternal mRNA degradation-related genes during the minor stage, suggesting differences in regulatory mechanisms and targets of mRNA degradation between minor and major ZGA (Figure 6D and Appendix A).

We performed clustering on the common set of maternal mRNAs with nonsignificant differences between in vivo and in vitro embryos to explore the impact of differences in maternal mRNA degradation. Category C5 was the only gene continuously degraded, selected for subsequent analysis (Figure 6E). In this category, the number of downregulated genes during the minor ZGA stage was significantly lower in IVF and PA than in IVV, and the intersections among the three embryo types differed (Appendix A), potentially due to the difference in mRNA binding proteins that recruit mRNA decay-related complexes (Figure 6F). In the expression trajectory analysis of four mRNA binding-related genes, *CPEB1* consistently exhibited significantly higher expression levels in in vivo conditions compared to in vitro during the minor ZGA stage. Notably, *ZFP36L2* in PA embryos did not show the timely compensation observed in IVF embryos (Figure 6F). The enrichment results of the independent intersection of C5 and IVV significantly downregulated genes during the minor ZGA, showing their correlation with negative regulation of phosphatase activity, material transporting, and cell junction (Appendix A), and further confirming, in relative GO terms (regulation of phosphatase activity and cell junction), that GSVA showed the consistent degradation trend (Appendix A), suggesting a potential arrest in these directions through an incomplete mRNA decay.

Additionally, some key genes, such as *POU5F1* and *PRKAA1*, although showing some upregulation during the major ZGA in in vitro embryos, rapidly degraded thereafter (Figure 1D and Figure 3E). We conducted a PPI analysis of significantly upregulated genes in IVV embryos during the 2–4-cell stage (Appendix A). Extracting the top five genes with the highest degree as core genes, we analyzed their trajectories and found that the mRNA degradation of these core genes was particularly pronounced in PA embryos (Appendix A). The heatmap of genes related to maintaining mRNA stability showed that genes related to mRNA decay, and genes related to mRNA stability form distinct categorizations into two groups at the minor and major ZGA stages, and the expression levels of these genes in in vitro embryos are far less abundant compared to in vivo embryos (Appendix A).

### 3.5. Expression Pattern of Mitochondrial Genes

During the early cleavage stages, embryonic development from the oocyte to the compact morula stages heavily relies on mitochondrial energy metabolism, which constitutes a relatively independent segment. At the beginning of our study, we observed significant differences in the percentage of mitochondrial (MT) gene expression among the three types of embryos (IVV, IVF, and PA) (Figure 1C), and IVV embryos exhibited a distinctive and independent enrichment of MT gene regulation at the major ZGA compared to in vitro embryos (Figure 5A,B). Of the 37 porcine MT genes listed in the Resembl database, only approximately 40% (15 genes) were effectively expressed, with the remainder primarily consisting of various tRNAs (Figure 7A). Notably, distinct expression and trajectory patterns between IVV and in vitro embryos were observed (Figure 7A). Therefore, a detailed investigation and analysis of these genes are crucial to fully understand their role in our study.

The 15 mitochondrially encoded protein-coding genes are involved in various oxidative phosphorylation processes (Figure 7C). The ND family genes (*ND1/2/3/4/4 L/5/6*) are core subunits of mitochondrial membrane respiratory chain complex I, while *COX*, *CYTB*, and *ATP* family genes constitute subunits of other complexes in the respiratory chain. 

To better identify and describe these differences, we characterized the expression trajectories of MT genes (Figure 7B) and initially divided them into two groups based on unsupervised clustering (Figure 7A). Most genes in Cluster 1 showed upregulated expression during the ZGA process, particularly from the two-cell to the four-cell stage, aligning with the importance of energy metabolism at this time, similar across all embryos. However, they exhibited rough expression levels at all stages in in vitro embryos. Remarkably, *ATP8* in Cluster 2 showed significant heterogeneity within the same group, with markedly higher expression in IVV embryos (Figure 7B,D). Here, only at the corresponding eight-cell stage did the expression levels in IVF embryos approach those in IVV embryos (Figure 7B,D).

Although most MT genes in IVV did not significantly diverge in expression from in vitro embryos during the major ZGA, Result 3 indicates that during this period, IVV underwent an independent enrichment process related to MT gene regulation (Figure 5A,B). Therefore, we further conducted an exploration of upstream/downstream genes of MT. The result showed that three transcription regulation-related genes and nucleosomal components associated with the proteome were significantly higher in IVV embryos during the major ZGA than in in vitro embryos (Appendix A).

## 4. Discussion

Embryo development, beginning with fertilization and proceeding through early stages, is regulated by an MTZ event. This transition shifts developmental control from maternal mRNA and proteins in the oocyte to the zygotic genome. While IVF and PA are crucial for techniques such as gene editing, they exhibit developmental deficiencies compared to IVV embryos, likely due to incomplete ZGA during MTZ in vitro. This study integrated single-cell transcriptomic data from three embryonic stages (oocyte to 8-cell) across swine oocytes and minor and major ZGA phases, providing insight into transcriptomic differences traceable to maternal mRNA.

From the oocyte to the eight-cell stage, maternal mRNA degradation surpasses zygotic gene expression. During this process, the number of differentially expressed genes of PA embryos, which have the most significant developmental defects due to the absence of paternal chromosomes, despite exhibiting more upregulated genes from the oocyte to the four-cell stages than IVF, is the lowest at each stage across all embryos, indicating limited developmental potential [46]. More differences were observed in the number of features but in counts, suggesting that transcriptional activity may not be the sole difference between in vivo and in vitro embryos but rather expression regulation. Although the similarity in significantly upregulated genes between stages is low across all three embryos, overall transcriptomic differences are smaller between in vitro embryos than between in vitro and in vivo embryos, suggesting maternal mRNA variances as a key factor in embryonic development [47].

In oocyte maturation, the IVM technique can significantly influence mRNA production, showing notable transcriptomic differences compared to in vivo maturation. Our analysis of oocyte discrepancies reveals divergences in several aspects. First, critical transcription factors for early embryonic development are stored in oocytes [19], with ELK1 [48], ELK3, ELK4, and GABPA present in significant quantities across all embryos, possibly explaining minimal differences in this aspect. However, only *POU5F1* was a maternal mRNA that maintained a relatively high expression level in IVV among all three embryos, suggesting its involvement in the whole process of MTZ, particularly postzygotic genome reprogramming in the 1–2-cell stage, along with pluripotency factors related to chromatin remodeling, such as *NANOG*, *c-MYC*, *SOX2*, and *KLF4*. *SOX2*, as a crucial major ZGA gene, lacks effective expression in vitro [49]. Second, significantly up- and downregulated genes show deep involvement in GO terms for gene expression regulation, involving mRNA processing, such as mRNA alternative splicing (AS), a crucial process that changes genomic instructions into functional proteins, playing a critical role in the regulation of gene expression in early embryos [50], and chromatin 3D structural adjustments based on DNA and histone modifications. Vital differential genes involved in these chromatin regulations have a close association with each other rather than random, sporadic aberrant expression, such as extraordinarily high-fold change genes, and they regulate gene expression in a stronger cooperative way, such as a high in vitro expression of *EZH2*, *SUZ12* (*PRC2* subunits) [51], and *SETD1B* [52], which silence genes via H3K27 trimethylation together. 

Early embryonic development is primarily controlled by maternal mRNA, and differences in maternal mRNA gene regulation likely directly impact the embryonic expression of further regulation of genes during the minor ZGA phase, leading to a cascading effect on subsequent major ZGA. First, the key differentially expressed genes in maternal mRNA involve different chromatin remodeling complexes and various chromatin modifications, with a very small intersection evident in the first wave of minor ZGA. In the entire minor ZGA stage, the upregulated genes in all three types of embryos are involved in chromatin remodeling; among these, IVV embryos, compared to in vitro embryos, have more participating genes and a richer variety of regulation in H2A ubiquitination [53] and H4 acetylation, which may control nucleosome accessibility of promoters prior to ZGA [54]. In the major ZGA stage from the 2- to 4-cell stages, IVV further participates in the acetylation regulation of histones independently. Additionally, overall, during the minor ZGA period, the gene set related to DNA methylation-dependent heterochromatin [55] formation was more abundantly expressed in in vitro embryos, while the gene set for histone acetylation was the opposite in vivo. These differences, inherited to some extent from the oocyte stage, also cause all three embryos to have extremely limited similarities in upregulated genes in subsequent cellular stages, whether in minor ZGA or major ZGA.

During MTZ, the orderly degradation of maternal mRNA is crucial. Our research indicates that among maternal mRNAs with comparable expression levels in both in vivo and in vitro environments, a specific category related to the negative regulation of phosphoprotein phosphatases (PPP) and cell junction [56] begins to degrade during the minor ZGA phase in vivo. PPPs, along with protein kinases, maintain a dynamic balance essential for controlling the cell division cycle’s various stages. This balance, critical for setting the phosphorylation state of proteins, is particularly vital during mitosis [57,58,59]. However, these PPP-related mRNAs are not effectively degraded in the in vitro environment, highlighting a key difference in embryonic development under these two conditions. This could be due to variations in the expression of genes involved in mRNA degradation, particularly those binding to mRNAs, such as CPEB1 [60]. Additionally, it is important to note that within the *CNOT* (*CCR4-NOT* transcription complex) subunits—specifically *CNOT7*, *CNOT6L*, and *CNOT11*, compared to *CNOT1/2/3/4/6/8/9/10* from minor ZGA to major ZGA—similar functions may be undertaken by different subunits at various stages, such as *CNOT6* and *CNOT8* [61]. Key genes during ZGA exhibited a more rapidly degradation in vitro, partly due to the differential genes related to mRNA stabilization through incomplete major ZGA in in vitro embryos. This phenomenon may be partly attributed to the differential expression of mRNA stabilizing genes, resulting from incomplete major ZGA in in vitro embryos.

Early embryonic development involves multiple biological activities, such as cell cycles, signaling pathways, chromatin remodeling, and biosynthesis, all requiring energy. Mitochondria are the primary energy source before embryo implantation. Porcine embryos depend mainly on lipid utilization for energy and are capable of developing with only lipids as a substrate [32]. Differential functional enrichment shows that in vivo embryos, unlike in vitro embryos, start upregulating lipid metabolism and energy-related pathways, such as AMPK signaling [62], from the minor ZGA stage. From minor to major ZGA in IVV, upregulated genes are involved in processes such as oxidative phosphorylation and thermogenesis. During major ZGA, metabolic processes related to ATP synthesis and mitochondrial regulation become prominent, with increased focus on these aspects from the two to eight-cell stages. However, substantial lipid metabolism, including PPAR pathway activation [63], is postponed to the 4–8-cell stage. Lipid metabolism and histone acetylation are closely linked, possibly explaining the higher histone acetylation activity in IVV [64]. Mitochondrial-encoded genes are significantly upregulated in both minor and major ZGA stages, with in vitro embryos showing higher expression of many mitochondrial genes compared to in vivo embryos at various stages. This suggests that in vitro conditions might enhance mitochondrial gene expression. Nonetheless, proteome comparisons should reveal higher levels in IVV. Meanwhile, *ATP8*, encoding a mitochondrial ATP synthase subunit [65], shows no effective expression in PA and is expressed until the eight-cell stage in IVF. This gene’s significant heterogeneity within the group remained unclear. The energy metabolism-related differences indicate that in vitro embryos fail to initiate energy metabolism in the MTZ of early embryos, especially during major ZGA.

This study, while shedding light on embryonic development differences using single-cell RNA sequencing, has its limitations. The exclusive focus on transcriptomic data might overlook some aspects of chromatin remodeling and energy metabolism. Incorporating epigenomic or metabolomic analyses in future research could provide a more holistic understanding. Additionally, examining beyond the early ZGA stages could reveal further insights into developmental processes.

## 5. Conclusions

In conclusion, our study discovered that the MTZ process exhibits multidirectional differences in IVV, IVF, and PA embryos. We found that the functional discrepancies in ZGA are primarily concentrated in chromatin regulation and energy metabolism. Moreover, we pinpointed key hub genes within these processes. This provides crucial insights and resources for enhancing assisted reproductive techniques in early in vitro embryo development.

## Figures and Tables

**Figure 1 animals-14-00465-f001:**
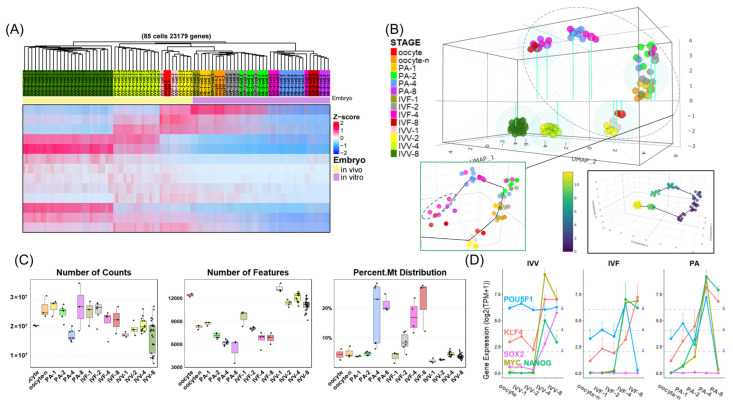
Comparative characteristic analysis of single-cell RNA-sequencing for IVV, IVF, and PA embryos from oocyte-8 cell stage. (**A**) Hierarchical clustering (top) and expression heatmap (bottom) of all genes from the remaining 85 samples after selection at the oocyte-8 cell stage for IVV, IVF, and PA embryos. (**B**) Top: 3D UMAP of all-gene expression for all 94 samples at the oocyte–8-cell stage of the IVV, IVF, and PA embryos. Bottom-left: A zoomed-in portion of the above figure showing oocyte–8-cell-stage in vitro embryos and oocyte–2-cell-stage in vivo embryos with pseudotime trajectories; blue circles highlight three 4-cell IVF samples falling within the 8-cell range. Bottom-right: the top image with pseudotime trajectories and samples colored according to pseudotime. In subsequent figures, the terms ‘Oocyte-n’ and ‘oocyte’ consistently refer to the same developmental stages, with ‘Oocyte-n’ denoting oocytes matured in vitro and ‘oocyte’ indicating those matured in vivo. (**C**) From left to right: box plots showing total gene expression, number of different genes, and percentage of mitochondrial-encoded gene expression for the 85 samples at oocyte–8-cell-stage of the three types of embryos. (**D**) From left to right: expression trajectories of *POU5F1*, *SOX2*, *KLF4*, *NANOG*, and *MYC* from the oocyte to the 8-cell stages in the IVV, IVF, and PA embryos, respectively.

**Figure 2 animals-14-00465-f002:**
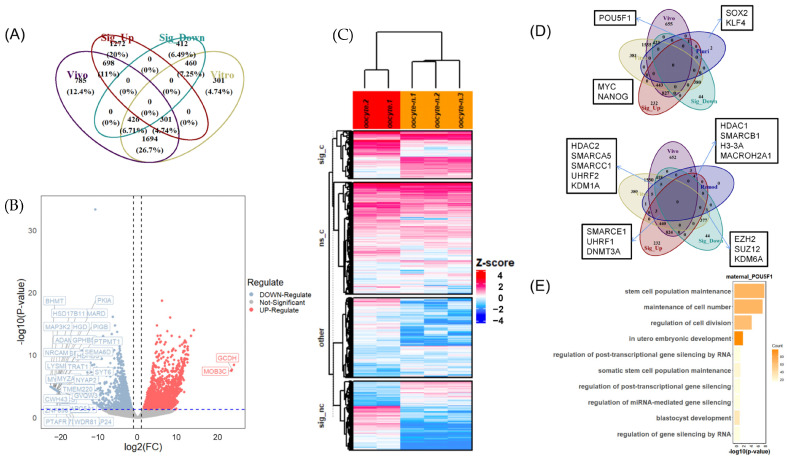
Maternal mRNA set definition and differential analysis between in vivo and in vitro matured oocytes. (**A**) Venn diagram showing the interrelationships among four gene sets: ‘Vivo’ represents the gene set in IVV oocytes with FPKM > 10; similarly, ‘Vitro’ pertains to the gene set in IVF oocytes. ‘Sig_Up’ denotes the gene set significantly upregulated in in vivo oocytes compared to in vitro oocytes, and, conversely, ‘Sig_Down’ refers to genes that are significantly downregulated. (**B**) Volcano plot annotated with the 10 genes that exhibit the highest and the lowest log2-fold changes, respectively. (**C**) Heatmap of gene expression across all oocyte samples, segmented according to the significance of gene expression differences. ‘sig_c’ denotes the genes with significant differential expression found in the common intersection between the Vivo and Vitro groups. ‘sig_nc’ represents genes with significant differential expression outside of the common intersection. ‘nc_s’ indicates the nonsignificant portion within the common intersection, and ‘other’ refers to the remaining genes not categorized in the aforementioned groups. It should be noted that this heatmap is globally normalized in contrast to other heatmaps in the study, which are normalized by row. (**D**) Top: Venn diagram illustrating the overlap between the four gene sets from diagram A and the pluripotency factor set. Bottom: Venn diagram depicting the intersection between the four gene sets from diagram A and a subset of genes related to chromatin remodeling. (**E**) GO enrichment results for *POU5F1* in the maternal mRNA milieu of in vivo matured oocytes.

**Figure 3 animals-14-00465-f003:**
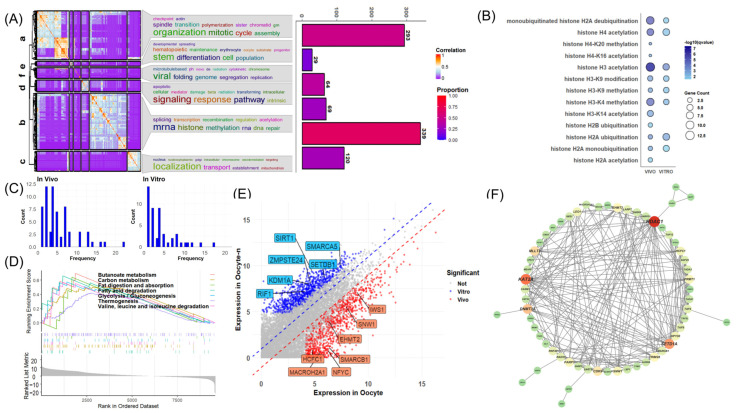
Enrichment analysis of differential maternal mRNA and comprehensive gene screening. (**A**) Left: GO terms derived from the enrichment analysis of the in vivo maternal mRNA set were chosen based on their correlation with significantly expressed in vivo genes. Subsequently, these terms were categorized into six distinct clusters (labeled a–f), and each cluster is depicted through a word cloud, highlighting the prevalence of various biological processes. Right: a bar chart displays the number of GO terms and the proportion of associated genes within each category. (**B**) Bubble chart showing GO terms related to histone modifications from category C for in vivo matured oocytes and category B for in vitro matured oocytes, derived from GO enrichment and clustering of differentially expressed genes. (**C**) From left to right: distribution charts illustrate the number of GO terms associated with each gene in the ‘remod’ collections in in vivo matured oocytes and in vitro matured oocytes, respectively. (**D**) GSEA plot showing important KEGG terms related to energy metabolism in in vivo matured oocytes compared to in vitro, with a threshold of NES > 1 and FDR < 0.25. (**E**) Scatter plot illustrating the correlation between the expression levels (log2(TPM + 1)) of genes in in vivo matured oocytes and in vitro matured oocytes. Genes with over 8 and 10 associated GO terms are labeled, respectively. (**F**) The protein–protein interaction (PPI) plot of ‘remod’ collections in in vivo matured oocytes.

**Figure 4 animals-14-00465-f004:**
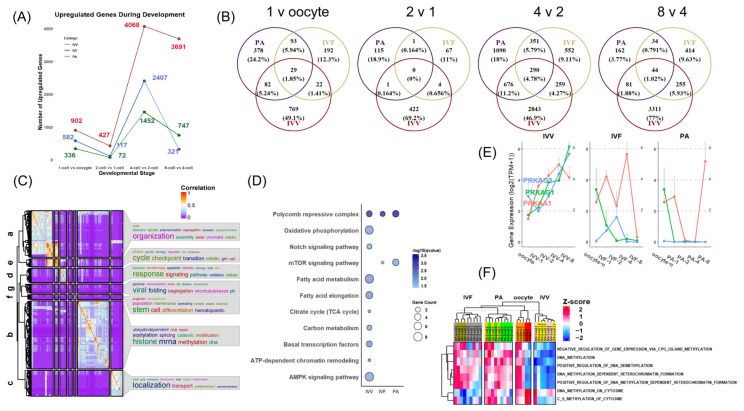
Differential analysis of adjacent stages from 1–8-cell stages and enrichment analysis of minor ZGA. (**A**) Line chart depicting the number of significantly upregulated genes from IVV, IVF, and PA embryos based on differential analyses of adjacent cell stages spanning from oocyte to 8-cell stages (threshold: FDR < 0.05, log2FC > 1). (**B**) Venn diagram illustrating the overlapping of significantly upregulated genes from differential analysis of adjacent stages spanning from oocyte to 8-cell stages among IVV, IVF, and PA. (**C**) GO terms resulting from the enrichment analysis of a gene set that combines the minor ZGA gene set of IVV and the maternal mRNA set from in vivo matured oocytes were selected based on their correlation with the significantly upregulated genes independent of PA and IVF. These terms were subsequently categorized into six distinct clusters (labeled a–f), and each cluster was represented by a word cloud highlighting the prominence of different biological processes. (**D**) Bubble chart depicting KEGG terms related to energy metabolism and chromatin remodeling of independently significantly upregulated genes during minor ZGA among IVV, IVF, and PA. (**E**) From left to right: expression trajectories of *PRKAG1*, *PRKAG2*, and *PRKAA1* from the oocyte–8-cell stages in the IVV, IVF, and PA embryos, respectively. The black line represents the average level. (**F**) GSVA of GO terms related to DNA methylation from the oocyte to 2-cell stages among IVV, IVF, and PA.

**Figure 5 animals-14-00465-f005:**
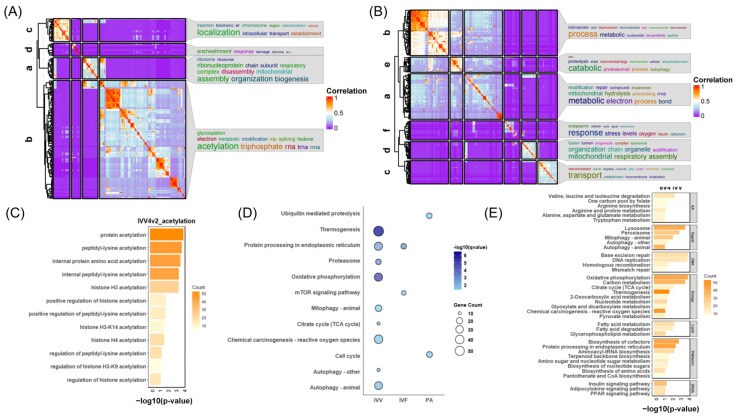
Enrichment analysis of major ZGA. (**A**) GO terms resulting from the enrichment analysis of independently, significantly upregulated genes during IVV major ZGA at the 2–4-cell stage, focusing on those terms that are distinct and independent of IVF and PA. These selected terms were then categorized into four distinct clusters (labeled a–d), and each cluster was visually represented as a word cloud, emphasizing the prominence of various biological processes. (**B**) GO terms resulting from the enrichment analysis of significantly upregulated genes during IVV major ZGA at the 4–8-cell stage. These selected terms were also categorized into six distinct clusters (labeled a–f), with each cluster visually depicted as a word cloud to highlight the diversity of biological processes. (**C**) The bar chart displays GO terms related to acetylation, which are part of category b in Panel (**A**). (**D**) Bubble chart depicting KEGG terms of significantly upregulated genes during major ZGA at the 2–4-cell stage among IVV, IVF, and PA. (**E**) The bar chart displays GO terms of significantly upregulated genes during major ZGA at the 4–8-cell stage in IVV.

**Figure 6 animals-14-00465-f006:**
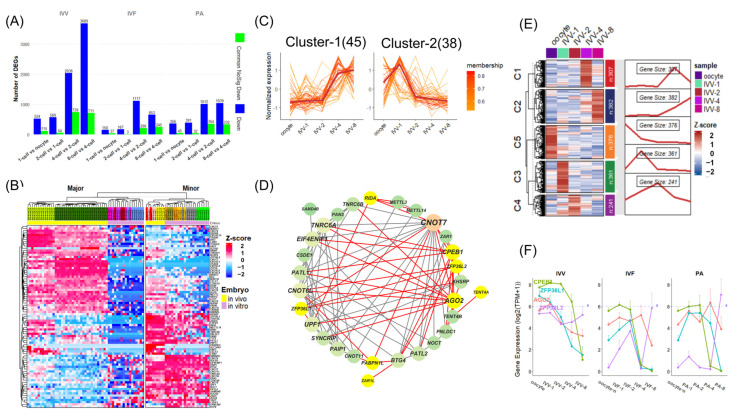
Differences in mRNA degradation in in vivo and in vitro embryos. (**A**) This panel shows the number of all significantly downregulated genes from oocytes to the 8-cell stage in each adjacent phase in blue, and the degradation of the non-significant-difference common set of maternal mRNAs in in vivo and in vitro embryos is represented in green. (**B**) A heatmap of gene expression for maternal mRNA degradation-related genes from the oocyte to the 8-cell stages in three types of embryos: IVV, IVF, and PA. (**C**) Clustering results for maternal mRNA degradation-related genes. (**D**) PPI network displaying the interactions among genes in Cluster 2 from Panel (**C**). The network is marked in bright yellow and red connections to denote genes that are significantly upregulated in in vivo matured oocytes compared to in vitro matured ones and their interactions. (**E**) From left to right: expression heatmaps and clustering results for the non-significant-difference common set of maternal mRNAs in in vivo and in vitro embryos, from the oocyte to the 8-cell stages. (**F**) From left to right: expression trajectories of *AGO2*, *KHSRP*, *ZFP36L1*, *ZFP36L2*, and *CPEB1* from the oocyte–8-cell stages in the IVV, IVF, and PA embryos.

**Figure 7 animals-14-00465-f007:**
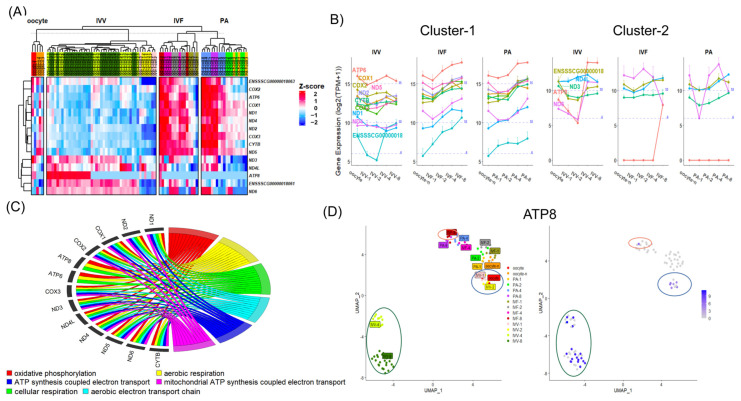
Differences in mitochondrial genes in IVV, IVF, and PA embryos. (**A**) A heatmap of gene expression for mitochondrial encoding genes from the oocyte to the 8-cell stages in three types of embryos: IVV, IVF, and PA. (**B**) Expression trajectories of mitochondrial encoding genes clustered by hierarchical clustering as shown in Panel (**A**). (**C**) Chord diagram displaying clustered GO enrichment results for mitochondrial encoding genes. (**D**) From left to right: UMAP of all samples, and expression feature maps for the *ATP8* gene across different stages and types of embryos.

## Data Availability

All in vivo embryo raw data of deep sequencing have been deposited in GEO (http://www.ncbi.nlm.nih.gov/geo/ (accessed on 25 March 2023)) as GSE139512.

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
