# Peer review of "Single-Cell RNA Sequencing Reveals Differences in Chromatin Remodeling and Energy Metabolism among In Vivo-Developed, In Vitro-Fertilized, and Parthenogenetically Activated Embryos from the Oocyte to 8-Cell Stages in Pigs"

_animals, 2024, doi:10.3390/ani14030465_

Round 1
Reviewer 1 Report
Comments and Suggestions for Authors
l Abstract and Summary:
Both the abstract and summary are informative but slightly lengthy. A concise presentation focusing on the key findings and their implications would be more effective.
l Clarity and Presentation of Figures:
The manuscript effectively utilizes analytic figures to support its findings. However, there are several issues that need attention. Some figures are not clearly referenced in the text, like in section 3.5 and the supplementary figure.4 . Additionally, there are minor errors in figure caption numbering and incomplete gene set names on certain Venn diagrams. Furthermore, some figure annotations lack detailed explanation, and the font size in several images could be increased for better visibility and reader experience. Addressing these discrepancies and enhancements will significantly improve clarity and aid the reader's understanding.
l Materials and Methods:
How many cells were used for sequencing should be described .
Line 209, Why not use the corrected P value?
Whether this study corrected for batch effects. Since IVV data come from your own sequencing, while IVF and PA data come from public databases, obvious differences can be found between IVV and IVF embryos. It is necessary to ensure that this difference is not caused by batch effects.
l Discussion of Limitations:
Incorporating a discussion in the manuscript about the benefits of integrating epigenomic or metabolomic data with transcriptomic analysis could significantly enhance the study’s depth. This addition would not only acknowledge the current limitations but also highlight how a multi-faceted approach could provide a more comprehensive understanding of chromatin remodeling and energy metabolism in future research.
Reviewer 2 Report
Comments and Suggestions for Authors
The manuscript contains interesting and new data concerning the molecular mechanism related to porcine embryo development under different fertilization conditions. The manuscript is interesting and can be published in Animals Journal but after corrections. The methods can be described in more detail, but the majority of the results are presented clearly and properly.
Specific comments:
The gene names should be in italics.
Section 2.1 – Has the experimental procedure got the number of approvals of the bioethics committee?
Were the Bama sows at similar age? The donor age can be an important experimental factor. Did you take it into account?
Can you show the exact number of analyzed embryos? The number of samples per each age group is not clear.
Whether a given period was represented by embryos from one or several mothers? Were the mothers related?
Line 186 - add the NGS sequencing parameters (cell cycles, SE or PE option)
Lines 208, 216 - which reference genome was used for GO and KEGG analyses?
Did you perform some validation steps as qPCR?
